# Ecological Benefits and Plant Landscape Creation in Urban Parks: A Study of Nanhu Park, Hefei, China

Shaowei Wu [1,2], Xiaojie Yao [1,2,*], Yinqi Qu [1,2] and Yawen Chen [1,2]

[1] School of Architecture and Planning, Anhui Jianzhu University, Hefei 230601, China; wsw786644135@163.com (S.W.); quinchi_77@163.com (Y.Q.); 13965267568@163.com (Y.C.)

[2] Anhui Institute of Land Spatial Planning and Ecology, Hefei 230601, China

* Correspondence: yaoxj661@sina.cn

**Abstract:** Plant landscape creation in urban parks is an important aspect of urban ecological construction under the goal of "carbon neutrality". In this study, the plant community of Nanhu Park in Hefei City was considered the research subject, and its tree species' composition and diameter at breast height (DBH) were analyzed. The ecological benefits of the park's green space were evaluated using the i-tree Eco model, and the carbon sequestration, runoff retention, air pollution removal, and oxygen production benefits were quantified as economic values and combined with the landscape effect evaluation method. The results show that Nanhu Park is rich in tree species types, with 5871 trees of 41 species in 23 families and 32 native species, among which three species of *Sapindus mukorossi*, *Eucommia ulmoides*, and *Triadica sebifera* accounted for 43.7% of the total number of trees. The dominant tree DBH was intermediate (7.6–15.2 cm). In Nanhu Park, the economic benefits were ordered as follows: carbon sequestration > runoff retention > air pollution removal > oxygen production benefits. The dominant tree species strongly contributed to the total ecological benefit of urban park green space; the ecological benefit of individual trees was not positively correlated with the number of tree species; native tree species had better ecological and landscape effects, while plant communities with growth changes and hierarchical depth of landscape were more popular. The analysis of ecological benefits and landscape evaluation of urban park green space provide a theoretical basis for enhancing the plant landscape, thus providing a case reference for promoting the construction of park green space in Hefei.

**Keywords:** carbon neutrality; ecological benefits; i-tree Eco; urban park green space

## 1. Introduction

Global climate change has brought on serious ecological challenges [1–3], and researchers have mainly elaborated on the current problems from various aspects such as nature (plant growth [4,5], land desertification [6,7]), national economy [8,9], agriculture [10–12]), and society (policy guidelines [13,14], urban construction [15,16], human health [17–20]). To cope with climate change and alleviate the deterioration of the environment, China has incorporated the "dual-carbon" policy into the construction of ecological civilization and the "14th Five-Year Plan" [21,22] to respond to the transformation and upgradation of contemporary society. The anthropogenic emissions will be reduced through energy conservation and emission reduction [23], ecological restoration [24], and other technical means, to maintain the carbon and oxygen balance, aiding carbon neutralization.

Research on carbon neutrality at home and abroad is abundant, including studies on the environmental Kuznets inverted U-shaped curve [25,26], which examines the relationship between environmental quality and the economy; the decoupling theory of the relationship between carbon emissions and economic growth [27]; the innovation of carbon capture, collection, and sequestration technology between energy, environment, public acceptance [28]; and other multi-disciplinary methodological and theoretical research. Some scholars in China have presented guidelines for the implementation of the

path and emission reduction methods for reaching the national carbon neutrality target through the energy structure [29], environmental governance [30,31] and other aspects. As the most densely populated and richly productive carbon emission key areas in the country, urban green spaces have become the only natural carbon sink space in the city, and urban park green spaces are the essential urban supporting construction land in the process of urban green development [32]. In terms of the benefits to human beings, urban parks with a high greening rate and a diverse range of tree species provide positive psychological guidance to human beings [33], reduce the occurrence of diseases [34], and regulate health conditions [35]. The diverse tree species constitute an interlocuter between human beings, society, and nature to provide multifaceted direct and indirect benefits for the modernized city, such as removing atmospheric pollution [36], reducing carbon emission [37], lowering the temperature of the city [38], and providing relief from human psychological stress [39]. It is also a concentrated expression of urban plant diversity and an important foundation for the stability of urban ecosystems. In recent years, people's perception of leisure and recreation has changed, with a growing inclination toward park construction that embraces a return to nature. There is an increasing focus on environmental aesthetics and plant landscape, emphasizing the need for a scientific and reasonable assessment of the visual landscape quality of parks [40]. This assessment is crucial for the protection and sustainable use of urban ecological resources, providing aesthetic value to human beings, as well as health services [41], through a reasonable combination of trees, shrubs, herbs and other elements in the plant landscape. The main statistical methods employed for the quantitative evaluation of people's visual perception of landscapes include analytic hierarchy process, scenic beauty evaluation (SBE), and semantic difference [42,43]. SBE is one of the most commonly used and effective methods in landscape evaluation. This method is implemented through on-site landscape photographs, which are subsequently scored by respondents through a questionnaire. SBE can quantify the psychological quantitative value of the evaluator's feelings or perceptions and objectively evaluate the actual aesthetic value of the landscape [44]. The results obtained through SBE are characterized by comprehensiveness and practical applicability [45].

How to maximize the ecosystem services of landscape plants, emphasize the function of plant carbon sequestration and oxygen release, and seek a balance between landscape and ecology under the guidance of the goal of "carbon neutrality" is a key issue to be considered in the future to achieve carbon neutrality in cities. This study involved the integration of two methods, the i-tree model and the SBE method, to evaluate the plant landscape in terms of its ecology and aesthetics. The objective of this study was to improve the ecosystem service capacity of park green spaces and to provide insights into the plant landscape considerations for the sustainable construction of urban ecology.

## 2. Materials and Methods

Based on the field census data of the trees in the park, the Nanhu Park in Hefei City was considered the study area. The structure and characteristics of the vegetation in the study area were analyzed using the i-tree Eco model [46]. The ecological benefits in terms of carbon storage and sequestration, runoff interception, removal of air pollution, and oxygen production were analyzed, and the park's landscape plant communities were evaluated via the weighted summation method of the evaluation of the scenery. The characteristics that received high scores were evaluated based on the landscape plant communities of the park through the weighted sum method of beauty degree evaluation. The characteristics of the landscape plant communities with high scores were analyzed. The enhancement methods for the landscape creation of urban park green spaces were presented in line with timely development needs to improve the ecosystem service capacity of urban park green spaces and the ecological environment of human habitats. Here, a case study for the construction of urban park green spaces system in Hefei City is presented.

### 2.1. Study Area

Hefei City (30°56′–32°33′ N, 116°40′–117°58′ E) is an area in the Anhui Province in East China with a typical subtropical monsoon climate, four distinct seasons, and an average annual temperature of 15–16 °C. The city is at the foot of Dashu Mountain in Hefei City, Anhui Province. Nanhu Park is located at the eastern foot of Dashu Mountain, covering an area of approximately 20 hm², built around the lake, with a water area of approximately 10 hm² (Figure 1). In 2017, the comprehensive management of the water environment of the lake was awarded the China Habitat Award for the environment. The site fully protects the original nursery land vegetation landform, with virtually no traces of artifacts, and retains the two original bird islands in the center of the lake to maintain the near-natural ecological environment to the maximum possible extent. The natural vegetation is in excellent condition, maintaining the original state of urban forest. The park is adjacent to Shufeng Bay Sports Park in the north and residential neighborhoods in the south and is a transitional area for human and nature contact.

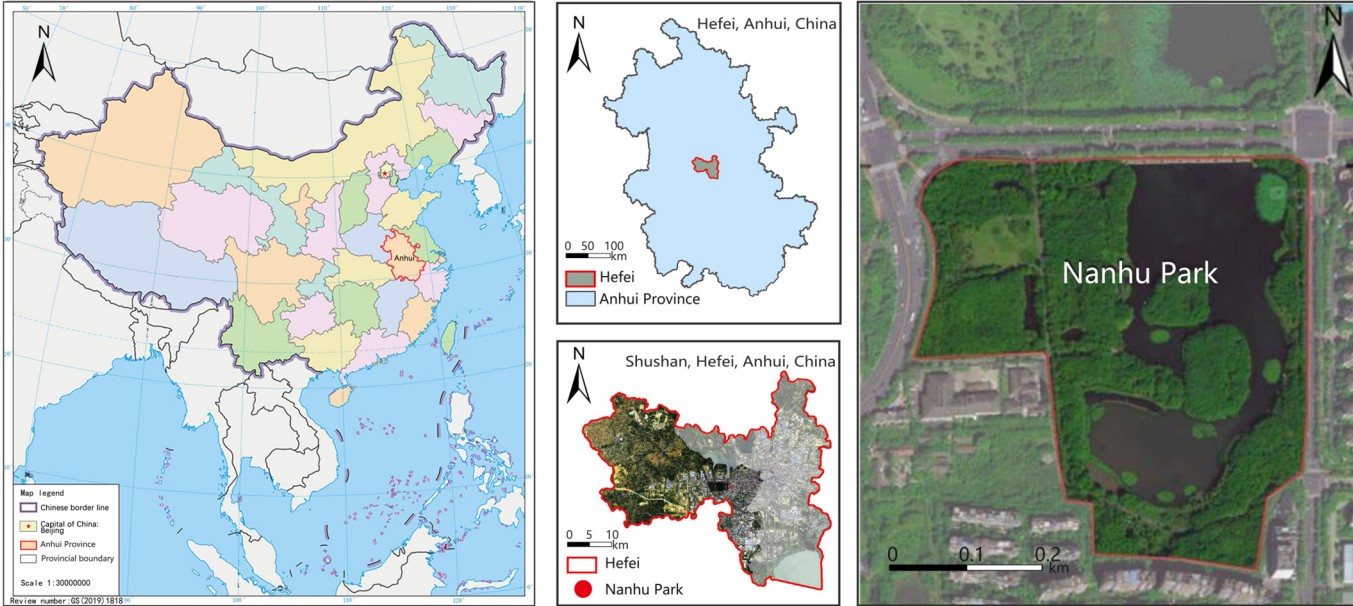

**Figure 1.** Geographical maps of Nanhu Park, Anhui Province, East China.

### 2.2. Dataset

Based on the i-tree Eco user manual and data collection form, a per-tree census was used to conduct a data survey from March 2021 to September 2022 on the garden trees eligible for measurement in this park. We obtained information on tree species, number, health status, under-branch height, crown width, crown deficiency rate, and light harvesting rate mainly during the growing period of trees from March to September. We measured the under-branch height of trees using measuring tape, used cell phone Google Maps software (http://www.gditu.net/) to locate the east–west and south–north orientation of trees, measured the vertical distance from the center of the tree trunk to the lateral branches in each orientation using measuring tape, and recorded the data on crown width. The tree diameter at breast height (DBH), tree height, and top height during the dormant period from September to March of the following year were calculated using a circumference ruler to measure the diameter of the tree trunk 1.3 m above the ground. The tree height was measured using a CGQ-1 altimeter aligned with the top of the tree crown and visually exploring surrounding buildings of known heights to assist in the correction. On 9 August 2022, at 9:00–11:00, images were collected using a 24.2-megapixel Sony camera (Sony(China) Co., Ltd.; Beijing, China) with a photo resolution of 300 DPI, in sufficient light, and from a human eye perspective, ensuring there was no human influence, to truly reflect

the ornamental nature of the tree assemblage. Seventeen clear and representative photos were selected from a large number of photos according to the evaluation object demand.

### 2.3. i-Tree Eco V6 Modeling

Currently, many scholars around the world use the i-tree series software developed by the U.S. Forest Service to analyze the tree structure and assess the ecosystem services of forest trees in the region [47,48]. Through the census of the study area or the list of sample plots, using the unified research data collection, together with the air quality information and annual rainfall information given by the weather station in the region, the analysis forms the data model for the standardization of the urban forest structural characteristics and urban ecosystem services [49]. The total carbon stock was calculated using the i-Tree Eco model, which controls the upper limit for calculating the weight of trees to prevent estimation errors. The total carbon sequestration was estimated by comparing the carbon stock of the current year with the carbon stock of the next year to estimate the carbon sequestration of trees in n + 1 years, and the absorption and fixation of carbon dioxide are calculated using the following formula:

$$\text{Carbon storage} = \text{Carbon storage factor} \times \text{Vegetation cover} \times \text{Area size} \tag{1}$$

Runoff retention was estimated using the Davis value (RD) according to the following equation:

$$RD = V \times C_{is} \times P \tag{2}$$

where $V$ is the total area of the study area, $C_{is}$ is the total effective impermeable surface rate, and $P$ is the average annual precipitation.

The formula for the amount of air pollution removed is as follows:

$$F = V_d \times C \tag{3}$$

where $F$ is the pollutant purification rate, $V_d$ is the deposition rate, and $C$ is the air pollutant concentration.

Total leaf area was estimated based on the urban tree leaf area regression equation, Eq:

$$Y = exp(0.631 + 0.2375H + 0.636D - 0.0123S_1) + 0.1824 \tag{4}$$

$$S_1 = \frac{\pi D(H + D)}{2} \tag{5}$$

where $Y$ is the total leaf area of a single plant, $H$ is the crown height, and $D$ is the crown width.

The air pollutant treatment fee (RMB 1.20/pollution equivalent) was determined based on the pollution tax amount for air pollutants specified in the Program on Applicable Tax Standards for Taxable Air Pollutants and Water Pollutants of Anhui Province Environmental Protection Tax implemented in 2018, and the electricity fee (RMB 0.565/kWh) was based on the Anhui Grid Sales Electricity Tariff Table issued by Anhui Development and Reform Commission in 2020 and the reference to the 2021 Swedish carbon tax rate to determine the economic value of carbon sequestration (0.85 RMB/kg); these values were used to correct the economic parameters within the software. The researched tree species were compared with the software database to match missing species with higher classified species or those with similar growth types.

### 2.4. Scenic Beauty Estimation (SBE)

The i-tree Eco reveals the ecological benefits of various plant communities in South Lake Park, while the SBE method can demonstrate plant landscape communities that not only offer significant ecological benefits but also receive high scores in aesthetics.

Plant landscapes are generally preferred over plant communities owing to their overall coordination, strong spatial sense, attractive forms, and high ornamental value of flowers and foliage [50,51]. This study used the SBE method [52] to evaluate the landscape of common tree communities in Nanhu Park. The 17 photos were randomly numbered and set in a slide show for 10 s. A total of 60 professional teachers and graduate students were invited to evaluate and score them using a 5-point system; one–five points from low to high represented very poor, poor, general, good, and excellent, respectively. After completing the statistics, the degree of beauty in the landscape was evaluated using a weighted summation of the results, employing the following formula:

$$S = 5 \times \sum_{i=1}^{5} \left( \frac{n_{ij}}{N} \times j \right) \tag{6}$$

where $S$ is the beauty value, $N$ is the total number of people evaluators, and $n_{ij}$ is the number of people scoring j for photo $i$.

## 3. Results

### 3.1. Tree Species Composition

In this survey, 5871 trees belonging to 41 species in 23 families in Nanhu Park were reported, with the species being dominated by *Sapindus saponaria*, *Eucommia ulmoides*, and *Triadica sebifera*. The DBH of 68.5% of the trees was between 7.6 and 15.2 cm, and only 1.1% of the trees had a DBH of more than 30.5 cm. A total of 1.1% of the trees had a DBH of >30.5 cm. Comparing the size of the DBH of the 10 tree species with the average age of the species at the same DBH in Hefei City, and based on the distribution of diameter at breast height, it was observed that the park possessed a large number of young trees, which were in the overall stage of stable growth, and could continue to contribute to the city's ecological benefits (Figures 2 and 3).

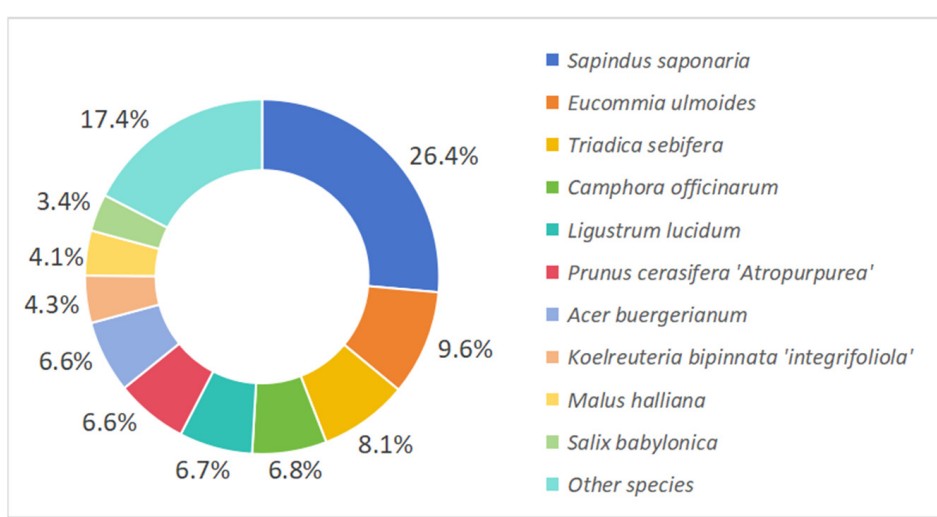

**Figure 2.** Percentage of tree species composition.

The increase in tree diversity can minimize the impact of species-specific insects or diseases, and the resistance and adaptability of native tree species can enrich the diversity of landscape plants and maintain the local ecological balance. According to the list of major native tree species in Anhui Province published by the Anhui Forestry Bureau in 2023, Nanhu Park comprised Hefei City's native tree species, including 19 species of *Sapindus saponaria*, *Eucommia ulmoides*, *Ligustrum lucidum*, *Acer buergerianum*, *Triadica sebifera*, and *Bischofia polycarpa*, which accounted for approximately 67% of the total species.

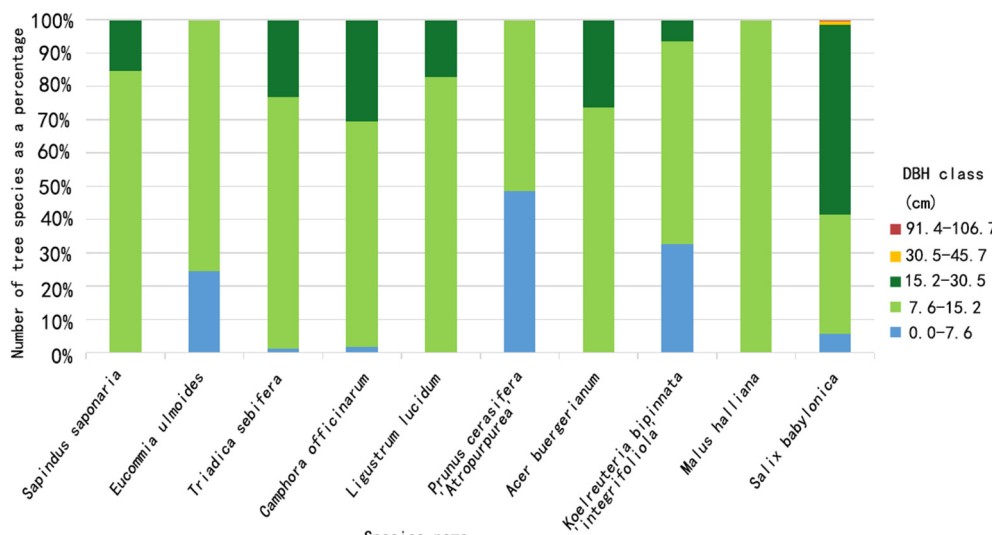

**Figure 3.** Percentage of diameter at breast height (DBH) of each tree species in Nanhu Park.

The tree cover of Nanhu Park was 8.70 hm², which provided 25.13 hm² of healthy leaf area. The leaf area was dominated by *Sapindus saponaria* and *Cinnamomum camphora*, and the percentage of number of species was added to the percentage of leaf area to obtain the 10 trees with high importance value (IV) (Table 1), which are the main part of the tree species composition of the Nanhu Park.

**Table 1.** Importance values of trees in Nanhu Park.

| Name of Tree Species | Percentage of Tree Species/% | Percentage of Leaf Area/% | Importance Value |
|---|---|---|---|
| *Sapindus saponaria* | 26.4 | 25.3 | 51.7 |
| *Cinnamomum camphora* | 6.8 | 13.0 | 19.9 |
| *Triadica sebifera* | 8.1 | 10.8 | 18.9 |
| *Ligustrum lucidum* | 6.7 | 9.7 | 16.4 |
| *Koelreuteria bipinnata'integrifoliola'* | 4.4 | 9.4 | 13.9 |
| *Acer buergerianum* | 6.6 | 7.0 | 13.5 |
| *Eucommia ulmoides* | 9.6 | 2.0 | 11.6 |
| *Prunus cerasifera'Atropurpurea'* | 6.6 | 1.4 | 8.0 |
| *Salix babylonica* | 3.4 | 3.2 | 6.6 |
| *Malus halliana* | 4.1 | 0.7 | 4.8 |

### 3.2. Analysis of Ecological Benefits

The eco-efficiency of urban parks is directly reflected in the absorption and sequestration capacity of planted trees for air pollutants, the carbon sequestration and the oxygen release capacity produced by trees through photosynthesis, as well as the interception and retention capacity of tree root systems and leaves for surface runoff and underground runoff. Importing the data of trees from Nanhu Park, Shufeng Bay, the simulation results are shown below (Figures 4 and 5):

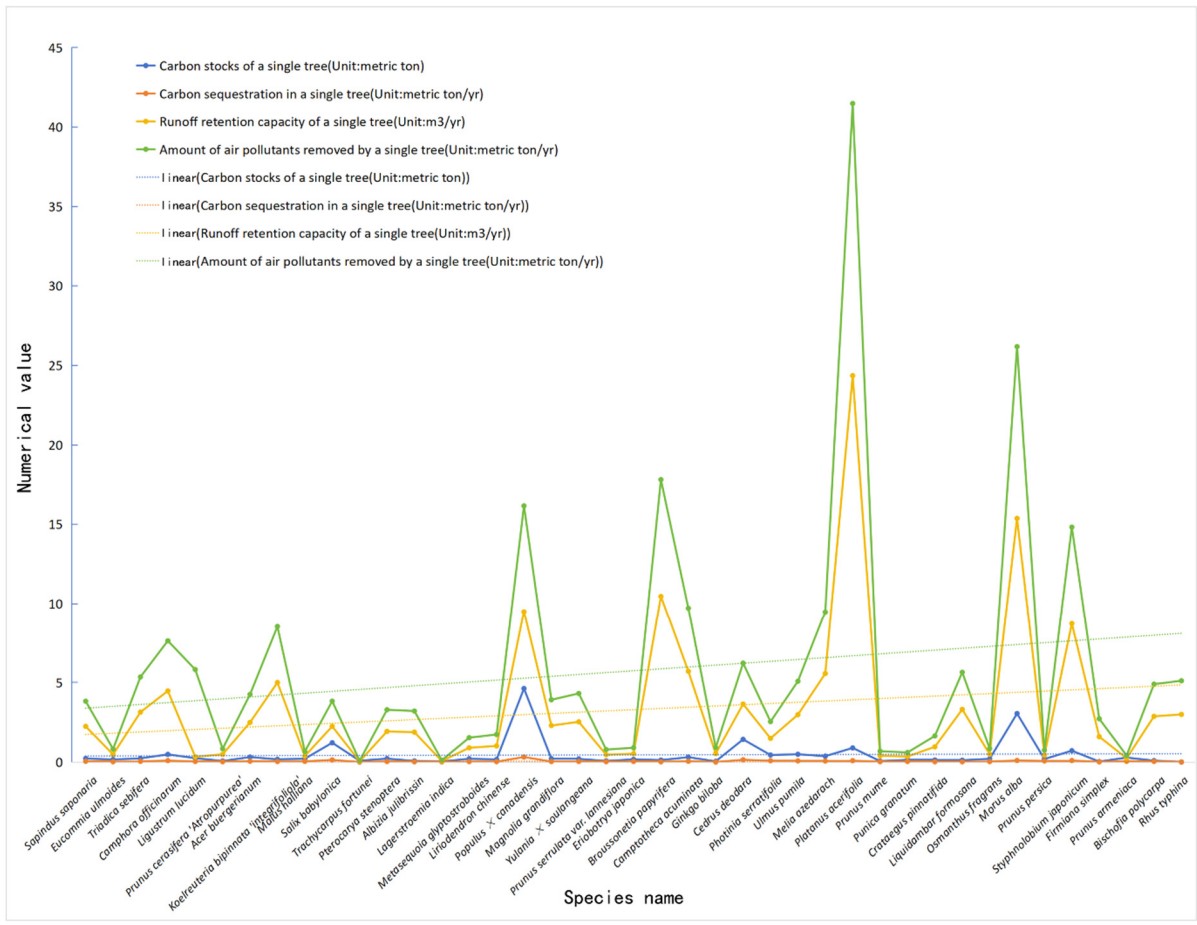

**Figure 4.** Comparison of total ecological benefits of different tree species.

1.  Carbon storage and sequestration

Trees absorb light energy and fix carbon dioxide in tree tissues through the process of carbon assimilation [53], and the amount of carbon sequestered per year varies with the size (crown width, DBH) and growth condition of the trees [54,55]. i-tree Eco calculated the total carbon sequestration of the trees in the Nanhu Park at 45.18 t per year, or approximately RMB 47,000, and the total storage at 245.34 t per year, or approximately RMB 254,000. Among the investigated plants, *Sapindus saponaria* stored and sequestered the most carbon, accounting for approximately 20.1% of the total storage and 25.9% of all sequestered carbon, while the largest amount of carbon stored in a single tree was observed in the *Populus* × *canadensis*, with an average of 0.63 t of carbon storage per tree per year. The largest amount of sequestered carbon was in the *Cedrus deodara*, with an average of 0.02 t of sequestered carbon per tree per year.

2.  Runoff retention

With increased urbanization, the use of large impervious surfaces in urban areas increases the amount of surface runoff, which may pollute and damage streams and wetlands, and excessive surface runoff can cause problems such as urban flooding [56]. Urban trees have a significant role in reducing surface runoff, with annual precipitation, leaf area and canopy size having a major impact on the amount of rainwater intercepted by trees. Trees in Nanhu Park help to reduce approximately 1876 m³ of the runoff per year, valued at approximately RMB 205,000, with *Platanus acerifolia* having the best retention effect, with a single plant reducing runoff by 3.34 m³ per year.

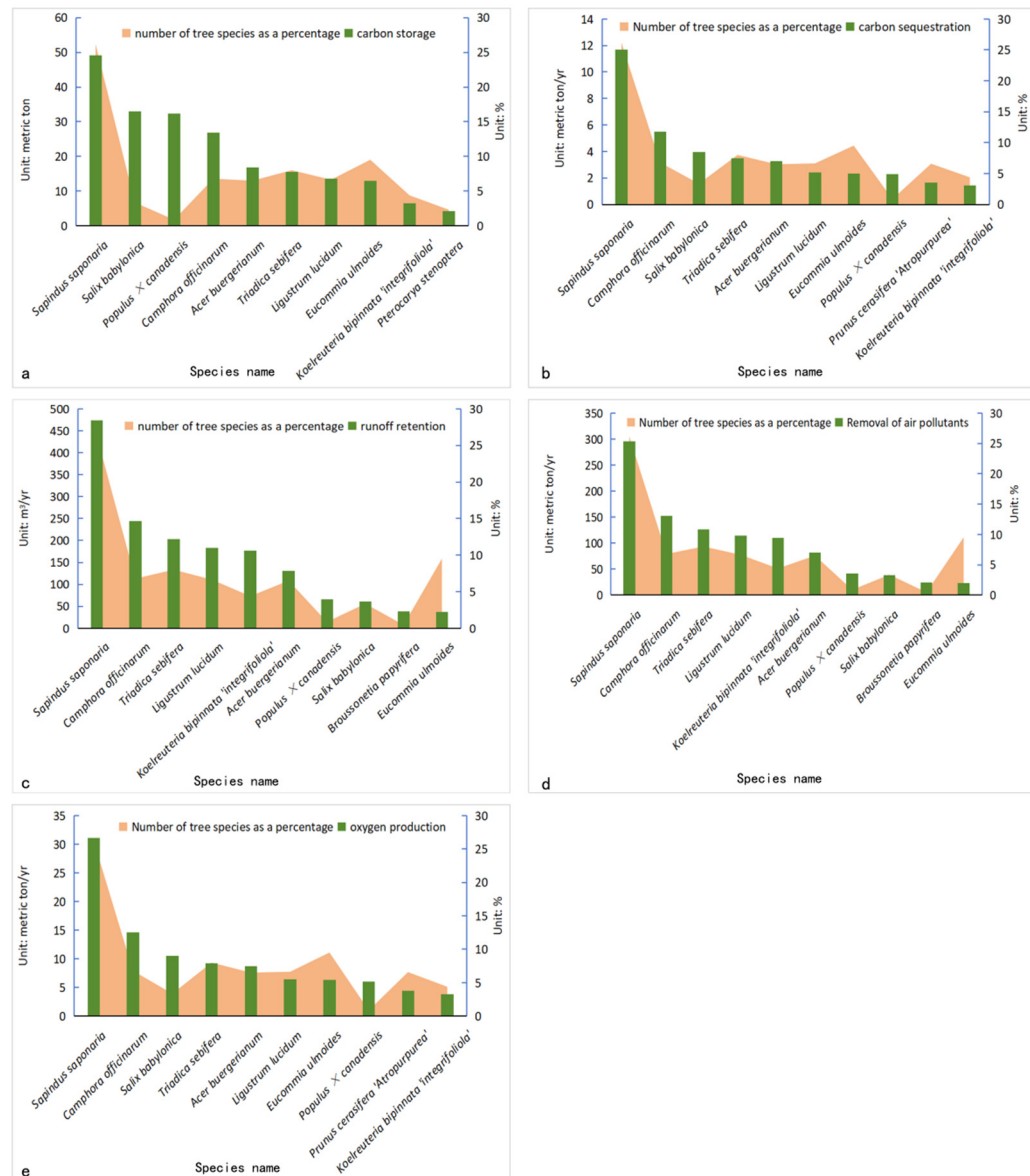

**Figure 5.** Top 10 tree species and percentage of:(**a**) carbon storage; (**b**) carbon sequestration; (**c**) runoff retention; (**d**) removal of pollutants; and (**e**) oxygen production.

3.　Removing air pollution

Trees play an important role in improving urban air quality. The amount of air pollutants removed by trees in Nanhu Park was calculated using the pollution and day data monitored by the Luo Gang weather station in 2019. Ozone had the largest amount of pollution removal (Figure 6), which was calculated according to the model that the trees in Nanhu Park could remove 1170 t of air pollution, which is approximately RMB 232,400. The ability to remove air pollution is related to tree species and crown size, and there is a small difference in the number of *Acer buergerianum* (382 plants), *Ligustrum lucidum* (390 plants) and *Camphora officinarum* (398 plants). The difference in the removal of air pollutants is obvious: *Acer buergerianum* (81.62 t/year) < *Ligustrum lucidum* (113.87 t/year) < *Camphora officinarum* (152.29 t/year).

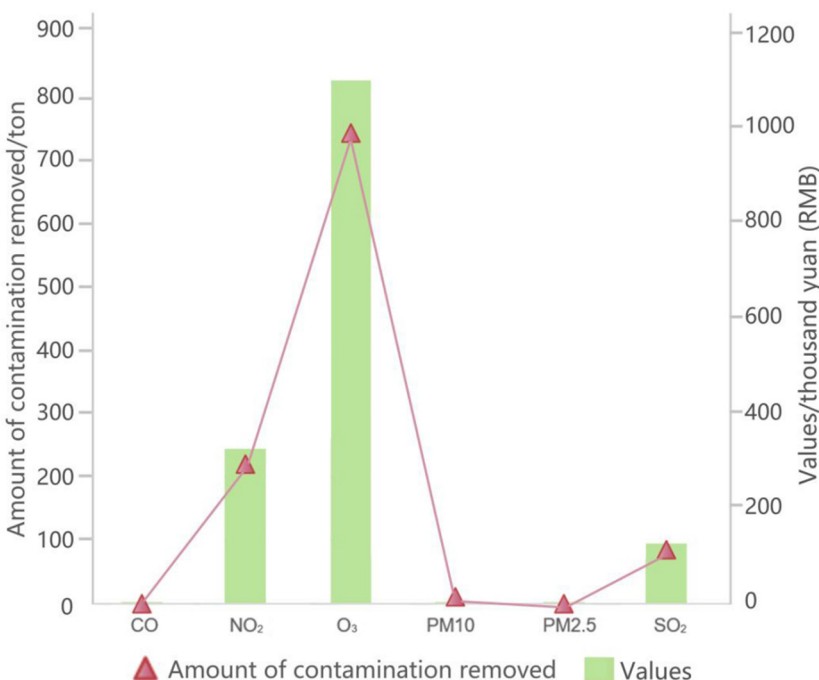

**Figure 6.** Amount and value of air pollution removed.

4. Oxygen production

Trees release oxygen by synthesizing organic matter through photosynthesis, and the amount of oxygen production is directly related to the amount of carbon sequestration. It is calculated that trees in Nanhu Park can produce 114 t of oxygen per year, and this study calculated the value of oxygen production at RMB 114,000 based on the recommended price of 1000 RMB/t in the 2008 specification for the assessment of forest ecosystem service functions.

*3.3. Landscape Evaluation*

The results of landscape evaluation were obtained through the weighted summation of the SBE method. The four groups of tree communities with the highest scores were Nos. 7, 4, 5, and 9 (Figure 7). The main species of the No. 7 community were *Camphora officinarum*, *Triadica sebifera*, with waterfront herbaceous plant *Pontederia cordata*, forming a semi-open space, and a sparse forest and grassland landscape by the lake, with a broader field of view, and a harmonious unity of the water and forest landscape, thus obtaining the highest score. The main tree species of the No. 4 community *were Platanus acerifolia* and *Prunus serrulata* var. *lannesiana*, forming a top-plane space; the height of the trees was in order, with flowers in the spring and leaves in autumn; the degree of beautiful scenery results was relatively high. The main tree species of the No. 5 community were *Prunus cerasifera* 'Atropurpurea', *Osmanthus fragrans*, *Camphora officinarum*, with *Hypericum monogynum*, enclosing a semi-open space. The colored leaf trees and evergreen trees formed a sharp contrast in leaf color, giving people a visual impact and making the rating relatively high. For the No. 9 community, the main tree species were *Populus × canadensis*, *Magnolia grandiflora* and *Koelreuteria bipinnata* 'integrifoliola', forming a vertical space at the entrance of the park. These trees have a certain advantage in the growth type, as they are tall and majestic; moreover, the road on both sides of the tree species, as well as the morphology and texture of the contrast, is impressive.

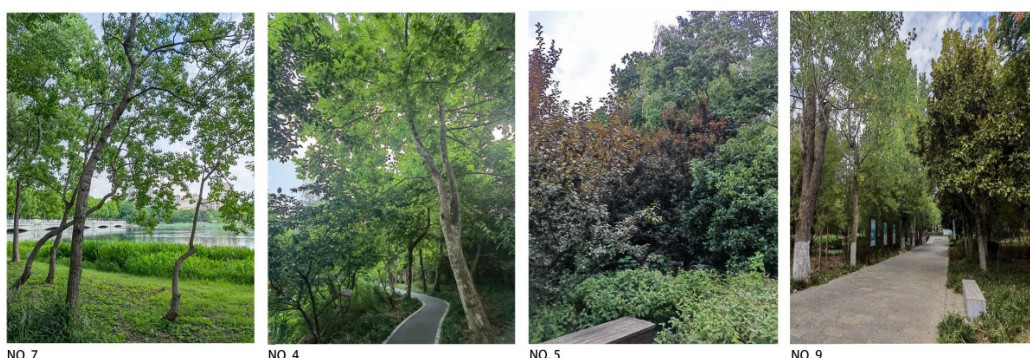

**Figure 7.** Map of the current status of the plant community.

## 4. Discussion

"Carbon neutrality" will be the long-term goal of sustainable urban development in the future. Under this goal, the selection of garden species should be based on the ecological and landscape considerations, wherein the selection of native tree species can produce better ecological and landscape spatial effects. The analysis based on the i-tree Eco model shows that among the economic benefits generated by urban park green spaces in carbon sequestration, air pollution removal, runoff retention, and oxygen production, these are ordered as carbon sequestration > runoff retention > air pollution removal > oxygen production benefits, which is consistent with the results of the studies conducted by Zhou Bening [57] and Wang Ying [58]. The benefits of single trees above the average value include 19 kinds of garden trees such as *Platanus acerifolia*, *Morus alba*, *Populus × canadensis*, *Broussonetia papyrifera*, *Koelreuteria bipinnata* "*integrifoliola*" (Figure 8). Plant landscape creation can make full use of the high eco-efficiency of the tree species, including the growth morphology, foliage texture, foliage and flower color differences, the relationship of the hierarchy between the trees, and the depth of the relationship between the combination of plants and the degree of integration of the surroundings to enhance the sense of aesthetics. The most appreciated of the higher rated communities are those with flower and leaf changes and depth of scene, indicating that people are more sensitive to the color, taste, and spatial hierarchical distribution of plants. Visual perception can be beautified by strengthening the seasonal changes in plant color and spatial hierarchical relationships in the park. The original nursery land vegetation planting characteristics of the park are more obvious, retaining part of the ranks planted, and tree species single-planting area; however, the visual landscape effect is poor, considering the space needed for plant growth, combing planting density distribution, appropriate reduction in seedlings, interplanting of different types of small trees and shrubs, and forming ornamental levels.

The population ecological benefit of dominant tree species *Sapindus saponaria* accounted for the largest proportion, approximately 29%, but as most *Sapindus saponaria* are in the youthful stage of trees, the ecological benefit of single trees only accounted for 2%; compared with the same DBH, different tree species and the crown widths of trees lead to different levels of carbon sequestration and oxygen release. The benefits of the removal of atmospheric pollution are in the middle level, which indicates that the level of the single-plant ecological benefit is related to the species of the trees, the DBH, the crown width, and the health of the trees. This is similar to the results of the study by Pengpeng Liu [59]. Native tree species accounted for 67% of the total tree species and approximately 78% of the total ecological benefits. Native tree species such as *Salix babylonica*, *Triadica sebifera*, and *Liquidambar formosana* are easier to plant and manage, change the focus of the park's forest canopy line and points of landscape appreciation, and the trees are able to enrich the façade space and color through changes in canopy, topography, and seasonal phases; moreover, they have certain regional characteristics. The rich tree species in Nanhu Park can maintain the current ecological environment of the park, protect the natural ecological

cycle system, improve the quality of forest trees in the park, prevent natural disasters, and provide a high-quality, sustainable ecosystem for the future development of the park.

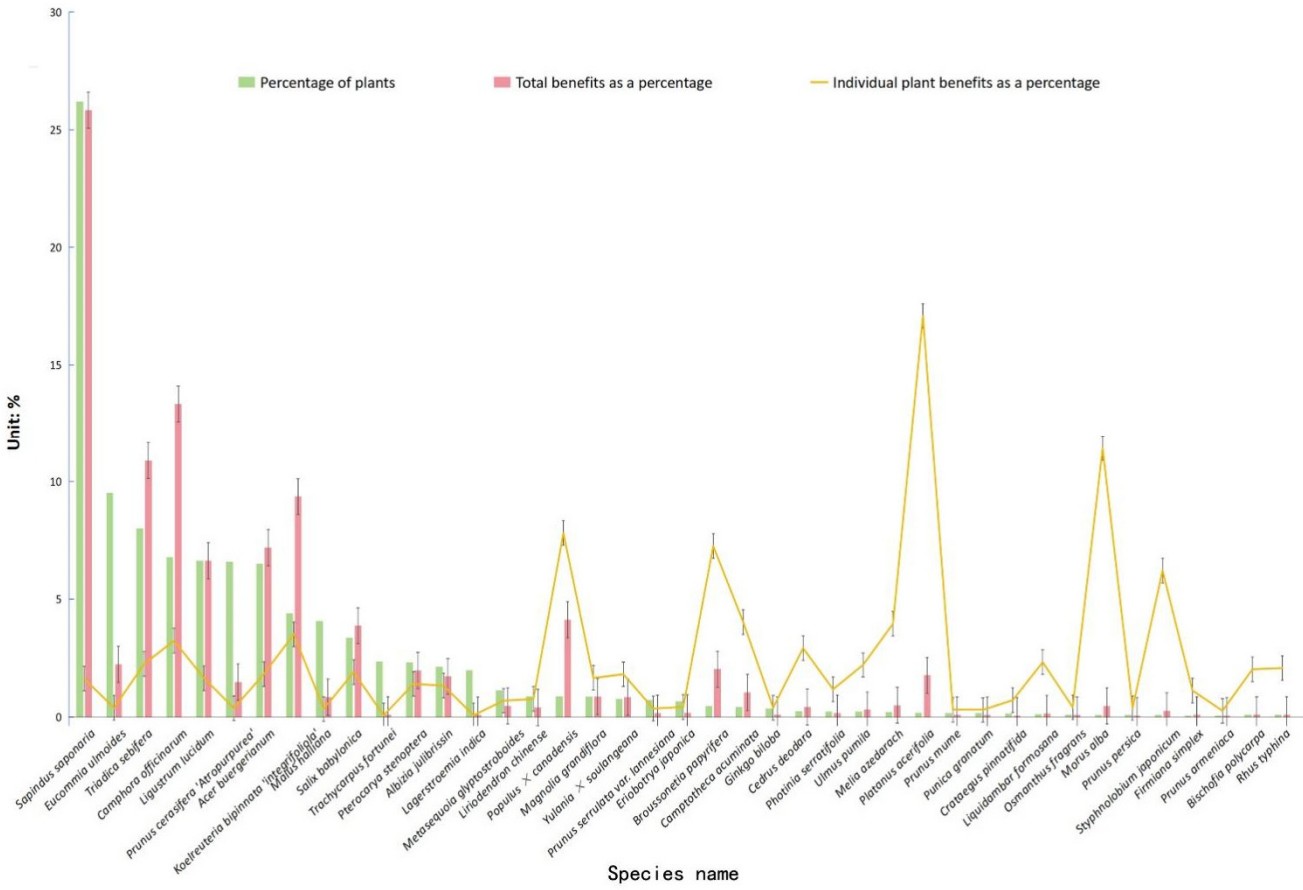

**Figure 8.** Comparison of ecological benefits of different tree species per tree.

The plant species and configuration of green space in urban parks vary based on the attributes of the nearby residents [60], emphasizing the importance of considering the relationship between the people in plant landscape design. Young people may prefer a combination of trees and herbaceous plants, creating an open and bright space suitable for picnics, socializing, sports, and other youth activities. The planting of evergreen trees and shrubs is more suitable for the elderly group, and fragrant trees such as pines and cypresses emit odors beneficial to their health. Trees and shrubs with brightly colored flowers and leaves are appealing to children, creating a lively and engaging atmosphere; however, while selecting plants, care should be taken to avoid poisonous and thorny plants. Urban park green space planting preferences evolve, considering the population attributes and based on the selection of dominant species [61]. The selection of dominant species is most beneficial when prioritizing the local ecological benefits of the native species. This not only benefits the overall stylistic tone of the park but also improves the overall ecological benefits, based on the planting design, zoning, reasonable configuration, targeted plant landscape design, and the sustainability of the plant landscape. Considering the sustainability of the plant landscape, dynamic characteristics of the subsequent design, optimization, and maintenance of plant landscapes in urban parks, as well as the nature-based transformation as a supplement to the optimization principle, the artificial plant landscapes can be organically integrated into the natural landscape in urban parks, based on the existing plant combinations, a bold design, and careful transformation.

This study focuses on the trees' ecological benefits and plant landscape aspects, proposing recommendations for the future construction of new urban park green spaces and the transformation of existing ones. The ecological benefits of trees can enhance landscape ef-

fects from the ecological and plant ornamental perspectives of the site space by prioritizing the allocation of strong native tree species, increasing their proportions, and enhancing the ecological benefits of the park. Matching big and small trees with high ornamental value can improve the species diversity of plant communities, enhance the landscape effect of the park, and contribute to the natural cycle system of plants. This aligns with the future direction of urban development guided by the goal of carbon neutrality.

## 5. Conclusions

(1) Nanhu Park is rich in tree species, with a total of 5871 trees of 41 species in 23 families, generally in a stable growth stage, and native tree species accounting for approximately 67% of the total number of tree species. When the same type of tree is planted on a certain scale, it becomes the dominant species in the park's plant community, thereby defining the botanical landscape of the entire park.

(2) There is a strong correlation between the dominant tree species in urban park green spaces and the total ecological benefits of urban park green spaces; the ecological benefits of individual trees are not positively correlated with the number of tree species but are more strongly correlated with plant growth characteristics, such as DBH, crown size, and health status; moreover, the park's carbon sequestration benefits outweigh its runoff, its retention benefits outweigh air pollution, and its removal benefits outweigh oxygen production benefits.

(3) The selection of tree species for urban park green spaces under the goal of carbon neutrality is based on native tree species with better ecological and landscape effects, and non-native tree species are used in moderation to enhance the overall landscape environment of the park. Plant landscapes have a high degree of correlation with people's vision, sense of smell and sense of space, and plant communities with floral and foliage variations and hierarchical depth of view are more popular.

(4) The optimization of green space in urban parks should be based on preserving the existing vegetation combinations, combining artificial and natural elements while maintaining the ecological balance of the landscape and the overall integrity of the park. Emphasis should be placed on overall coordination, ecological environmental protection, and the ornamental aspects of the park. This involves utilizing scenic features and enhancing the overall visual appeal of the landscape to provide visitors with a better touring experience, contributing to showcasing the city's green image and promoting ecological concepts.

(5) In various countries, cities pursuing low-carbon development need to calculate regional plant eco-efficiency and issue questionnaires to nearby residents in the construction and maintenance of urban parks. In practice, the ecological aspects and the nature of plants should be synthesized in design and the park plant communities should be configured to improve the potential of urban green development.

**Author Contributions:** Conceptualization, S.W.; Formal Analysis, S.W.; Funding Acquisition, X.Y.; Investigation, S.W., Y.Q. and Y.C.; Methodology, S.W.; Project Administration, X.Y.; Software, S.W. and Y.Q.; Supervision, X.Y.; Validation, Y.C.; Visualization, S.W., Y.Q. and Y.C.; Writing—Original Draft, S.W.; Writing—Review and Editing, X.Y. All authors have read and agreed to the published version of the manuscript.

**Funding:** This research was funded by the Anhui Provincial Natural Science Foundation, grant number 2308085ME183.

**Institutional Review Board Statement:** Not applicable.

**Informed Consent Statement:** Informed consent was obtained from all subjects involved in the study.

**Data Availability Statement:** Data supporting the results of this study are available from the first author, Shaowei Wu.

**Conflicts of Interest:** The authors declare no conflict of interest.

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
