# Peer review of "Ecological Benefits and Plant Landscape Creation in Urban Parks: A Study of Nanhu Park, Hefei, China"

_sustainability, doi:10.3390/su152416553_

Round 1
Reviewer 1 Report
Comments and Suggestions for Authors
Dear Authors,
The article is relevant, informative and good. However, I have some questions:
1) In the introduction there is no purpose of the article
2) In Figure 8, the fonts are small, I would like to see in detail
3) You can write a little bit in detail about the SBE method
4) Please decrypt the DBN
5) It would have been better if the report had recorded concrete results

Reviewer 2 Report
Comments and Suggestions for Authors
The value and usefulness of natural resources, plants and their ecological value, ecosystem services is an important scientific and practical issue of our time, and this article is an excellent choice of topic.
I suggest to move the last paragraph of the introduction to the Materials and methods chapter.
Scenic Beauty Estimation / Landscape effect is also very important for many reasons, but I found it a more complex topic just to analyse it the way how it is represented in the manuscript. I would suggest a more extensive literature review (landscape aesthetics, visual assessments, landscape architecture principles etc.) and a more elaborated evaluation method for this topic (e.g. the selection of view points and photos have to be consistent and clear), or it is also possible to drop EBS from this paper.
In the conclusions I suggest to add more content on how the results can be used for practical (maintenance, planning) purposes, and the relevance of the article for international readers.
Reviewer 3 Report
Comments and Suggestions for Authors
see Att.

Round 2
Reviewer 2 Report
Comments and Suggestions for Authors
I accept and I appreciate the changes made in the manuscript.
Author Response
Response to Reviewer
Manuscript ID: sustainability-2694793
Manuscript Title: Ecological Benefits and Plant Landscape Creation in Urban Parks: A Study of Nanhu Park, Hefei, China
Dear Editors and Reviewers:
Thank you very much for your suggestions on this manuscript and for agreeing with our first round of revisions. If you have any questions, please do not hesitate to contact me at any time.
Thank you and best regards.
Yours sincerely,
Shaowei Wu
Reviewer 3 Report
Comments and Suggestions for Authors
Based on the comments/suggestions from the reviewers, the author team has invested effort and time in carefully revising the original manuscript. At present, the quality of the second draft has basically reached a publishable level. After resolving some formatting issues (see below), it is acceptable to publication.
1) The entire paper should use “one name” instead of using "South Lake Park" in the title, "Nanhu Park" in the abstract, and confusing both in the main text. I suggest using "Nanhu Park".
2) In the author's work-unit column, some font styles are not standardized. It's Hefei, not “hefei”. The author's email address should also meet the requirements of MDPI.
3) The resolution of Figure 4 is not clear enough, the original image should be provided before publication.
4)It is recommended that the English language of the manuscript should meet the publishing level of MDPI.
